# Natural Product Auraptene Targets SLC7A11 for Degradation and Induces Hepatocellular Carcinoma Ferroptosis

**DOI:** 10.3390/antiox13081015

**Published:** 2024-08-20

**Authors:** Donglin Li, Yingping Li, Liangjie Chen, Chengchang Gao, Bolei Dai, Wenjia Yu, Haoying Yang, Junxiang Pi, Xueli Bian

**Affiliations:** 1The MOE Basic Research and Innovation Center for the Targeted Therapeutics of Solid Tumors, School of Basic Medical Sciences, Jiangxi Medical College, Nanchang University, Nanchang 330031, China; ldonglin0303@163.com (D.L.);; 2Shanxi Academy of Advanced Research and Innovation, Taiyuan 030032, China

**Keywords:** auraptene, HCC, ferroptosis, SLC7A11

## Abstract

The natural product auraptene can influence tumor cell proliferation and invasion, but its effect on hepatocellular carcinoma (HCC) cells is unknown. Here, we report that auraptene can exert anti-tumor effects in HCC cells via inhibition of cell proliferation and ferroptosis induction. Auraptene treatment induces total ROS and lipid ROS production in HCC cells to initiate ferroptosis. The cell death or cell growth inhibition of HCC cells induced by auraptene can be eliminated by the ROS scavenger NAC or GSH and ferroptosis inhibitor ferrostatin-1 or Deferoxamine Mesylate (DFO). Mechanistically, the key ferroptosis defense protein SLC7A11 is targeted for ubiquitin–proteasomal degradation by auraptene, resulting in ferroptosis of HCC cells. Importantly, low doses of auraptene can sensitize HCC cells to ferroptosis induced by RSL3 and cystine deprivation. These findings demonstrate a critical mechanism by which auraptene exhibits anti-HCC effects via ferroptosis induction and provides a possible therapeutic strategy for HCC by using auraptene or in combination with other ferroptosis inducers.

## 1. Introduction

Hepatocellular carcinoma (HCC), the main type of liver cancer, accounts for 75% to 85% of primary liver cancer cases and is the leading cause of cancer-related deaths worldwide [1,2,3]. Because of the limitations of strategies for HCC treatment, the overall survival rates of HCC patients are very low. Clinically, the multikinase inhibitor sorafenib is currently the effective first-line therapy drug used for late-stage HCC treatment [4]. Unfortunately, acquired resistance to sorafenib is increasing in patients with HCC and this problem needs to be addressed urgently [5,6,7]. Recently, sorafenib has been found to induce ferroptosis via inhibiting the cystine/glutamate antiporter subunit solute carrier family 7 member 11 (SLC7A11) [8,9]. Therefore, it is a promising therapeutic strategy for the treatment of HCC patients via using new compounds, especially natural products, in combination with sorafenib or any other medications.

Ferroptosis is a programmed cell death characterized by iron-dependent lipid peroxidation [10,11], which is tightly regulated by cellular iron and reactive oxygen species (ROS) levels. It is controlled by multiple cellular activities, including cell metabolism [12,13,14,15], redox homeostasis regulation [16], iron handling [17], and other signaling pathways [18]. There are two major mechanisms that suppress ferroptosis. One is mediated by the selenium-containing enzyme glutathione peroxidase 4 (GPX4) that catalyzes the reduction of phospholipid peroxides to lipid alcohols using glutathione (GSH) as a cofactor. The cystine/glutamate antiporter SLC7A11 is a major cystine transporter to mediate extracellular cystine uptake for glutathione biosynthesis [19]. Another pathway is mediated by cellular metabolic enzymes that can directly catalyze the production of radical-trapping antioxidants (RTA) [20,21], such as ferroptosis suppressor protein 1 (FSP1) [22,23] and dihydroorotate dehydrogenase (DHODH) [24]. FSP1 reduces ubiquinone to ubiquinol, which is an endogenous antioxidant metabolite that inhibits propagation of lipid peroxidation through radical trapping. DHODH is a mitochondrial enzyme that catalyzes the conversion of dihydroorotate to orotate with the reduction of CoQ10 to CoQ10H2, which acts as an RTA to defend lipid peroxidation in the mitochondria. Intriguingly, it was reported that tumor cells, especially metastatic or chemotherapy-resistant tumor cells, are sensitive to ferroptosis induction, thereby targeting tumor cell ferroptosis, and therefore this may be a promising strategy for cancer treatment [25,26].

Auraptene, the major coumarin derived from citrus plants and various citrus fruits such as Grapefruit and Poncirus trifoliata [27], is the most well-known and abundant coumarin found in nature which includes a geranyloxyl at its C-7 position. It has been reported that auraptene exerts anti-tumor, anti-inflammatory, antigenotoxic and neuroprotective activities [28]. Auraptene can function as an anti-tumor compound by inhibiting cell proliferation [29,30], migration and invasion [31,32], or inducing cell apoptosis [33] in a variety of cancer cells. Intriguingly, auraptene can exert cytotoxic effects via induction of ROS in the mouse colon cancer cell line CT-26 [33] and acute myeloid leukemia cell lines [30]. Excess cellular ROS levels can induce ferroptosis. However, whether auraptene has the effect of ferroptosis regulation in cancer cells and the detailed anti-tumor mechanism, especially in HCC cells, is largely unknown.

In this study, we demonstrated that the natural product auraptene targets SLC7A11 for ubiquitin–proteasome degradation, thereby exerting anti-tumor effects in HCC cells via ferroptosis induction.

## 2. Materials and Methods

### 2.1. Cell Lines and Cell Culture

HCC cell lines (HCCLM3 [34] and HLE [35]) were obtained from the Cell Bank of the Chinese Academy of Sciences. The cells were cultured in Dulbecco’s modified eagle medium (DMEM, Solarbio, Cat# 11965, Beijing, China) containing 10% fetal bovine serum (FBS, Excell Bio, FSP500, Shanghai, China). The cell lines were maintained at 37 °C with 5% CO_2_ in a humidified incubator. For cystine deprivation experiments, HCC cells were washed twice with phosphate-buffered saline (PBS) and then cultured in cystine-free DMEM containing 10% FBS for the indicated time.

### 2.2. Reagents and Antibodies

The reagents used in this study were Dimethyl sulfoxide (DMSO, Solarbio, Cat# D8371, Shanghai, China); 0.25% trypsin digestion solution (Servicebio, Cat# G4012, Wuhan, China); Cell lysis buffer (CST, Cat# 9803S, Danvers, MA, USA), PMSF (Wuhan Dingguo Biotechnology, Cat# 329-98-6, Wuhan, China); Auraptene (TargetMol, Cat# T4115, USA); N-acetyl-L-cysteine (NAC, MCE, Cat# HY-B0215, China); L-Glutathione reduced (GSH, MCE, Cat# HY-D0187, China); Ferrostatin-1 (Fer-1, TargetMol, Cat# T6500, USA); Deferoxamine Mesylate (DFO, TargetMol, Cat# T1637, Boston, MA, USA); Propidium Iodide (PI, TargetMol, Cat# T2130, USA); DCFH-DA (MCE, Cat# HY-D0940, China); C11-BODIPY 581/591 (MCE, Cat# HY-D1691, China); RAS-selective lethal small molecule 3 (RSL3, CSNpharm, Cat# CSN17581, Arlington Heights, IL, USA); Z-Leu-Leu-Leu-CHO (MG132, Biovision, Cat# 1791-5, San Francisco, CA, USA); Cycloheximide (CHX, MCE, Cat# HY-12320, China); Cell Counting Kit-8 (CCK-8, TargetMol, Cat# C0005, USA); GSH test kit (Servicebio, Cat# G4305-48T, China).

The antibodies used were as follows: Vinculin (Santa Cruz, Cat# sc-73614, Santa Cruz, CA, USA, used at 1:10,000); GPX4 (Proteintech, Cat# 67763-1-Ig, Wuhan, China, used at 1:2000); SLC7A11 (Proteintech, Cat# 18790-1-AP, China, used at 1:3000); FSP1 (Santa Cruz, Cat# sc-377120, USA, used at 1:3000); ACSL4 (Proteintech, Cat# 22401-1-AP, China, used at 1:8000); Flag (Sigma, Cat# 66008-3Ig, Burlington, MA, USA, used at 1:5000); HA (Santa Cruz, Cat# sc-7392, USA, used at 1:5000); HRP conjugated goat anti-mouse IgG (Thermo Fisher Scientific, Cat# 31437, Waltham, MA, USA, used at 1:8000) or goat anti-rabbit IgG secondary antibody (Thermo Fisher Scientific, Cat# 31460, USA, used at 1:8000).

### 2.3. Cell Viability Assay

Cell viability was assayed by Cell Counting Kit-8 (CCK-8, TargetMol, Cat# C0005, Boston, MA, USA). Briefly, HCCLM3 or HLE cells were plated into a 96-well plate at the density of 20,000 cells/well. After culture or treatment with the indicated periods, the culture medium was replaced with 200 μL CCK-8 reagents which were mixed with DMEM containing 5% FBS at a ratio of 1:10. The cells were incubated at 37 °C and 5% CO_2_ for 1 h. After incubation, the absorbance at 450 nm was measured with a microplate reader and cell growth curves were analyzed with GraphPad Prism 9.

### 2.4. Colony Formation Assay

Colony formation was performed as described previously [36]. Briefly, HCC cells were counted and plated into a 6-well plate and cultured at 37 °C and 5% CO_2_ for 24 h. The indicated compounds were added in the culture medium, and the cells were cultured for the indicated periods, then washed three times with PBS and fixed with paraformaldehyde (Solarbio, Cat# P1110, China) for 15 min at room temperature. After fixation, the cells were washed three times with PBS and then stained with crystal violet solution (0.1% crystal violet in 20% methanol) for 5 min. Finally, the stained cells were washed three times with PBS to remove free crystal violet and photographed.

### 2.5. Cell Death Analysis

Cell death rate was measured using Propidium Iodide (PI) staining. In brief, adherent cells and dead cells in the culture medium were collected and washed with PBS. The collected cells were then stained with 10 μg/mL PI for 15 min at room temperature in the dark. After staining, the cells were washed three times with PBS and resuspended with 500 μL PBS, followed by analysis with flow cytometry. FlowJo 10 and GraphPad Prism 9 software were used for analyzing and calculating the data.

### 2.6. ROS and Lipid ROS Levels Detection

The cellular ROS and lipid ROS levels were measured by flow cytometry. Briefly, cells were seeded in a 6-well plate and stimulated with the indicated compounds according to the experiment design. Then, the cell culture medium was removed and the probe DCFH-DA (for total ROS analysis) or C11-BODIPY 581/591 (for lipid ROS analysis) in 1 mL PBS was added into the plate and cultured for another 30 min in the cell incubator. DCFH-DA, a green fluorescent dye, is a probe for the detection of intracellular ROS (Ex/Em = 488/525 nm) with cell membrane permeability. C11-BODIPY 581/591 itself has red fluorescence (reduced; Ex = 581 nm, Em = 591 nm) and binds to lipids to produce green fluorescence (oxidized type; Ex = 500 nm, Em = 510 nm). After incubation, the cells were digested by 0.25% trypsin digestion solution (Servicebio, Cat# G4012, China), centrifuged at 1500 rpm at 4 °C for 2 min, washed twice with PBS and resuspended in PBS. The ROS and lipid ROS levels were detected by flow cytometry in the channel of FITC-H. The data were analyzed with FlowJo 10 and GraphPad Prism 9 software.

### 2.7. Western Blot

Cells were harvested and lysed with lysis buffer (CST, Cat# 9803S, USA) containing 1% protease inhibitor PMSF (Wuhan Dingguo Biotechnology, Cat# 329-98-6, China) at 4 °C for 15 min, then the lysed cells were transferred into a new 1.5 mL EP tube and centrifuged at 4 °C at 12,000 rpm for 15 min. The supernatants were collected into a new 1.5 mL EP tube followed by an addition of a 2 × SDS loading buffer at the ratio of 1:1 and boiled at 100 °C for 5 min. Finally, the lysate was used for SDS-PAGE analysis after electrophoresis. The proteins were transferred to the PVDF membrane (Millipore, Cat# 03010040001, USA) which was blocked in 5% skim milk (Solarbio, Cat# D8340, China) at room temperature for 1 h and then incubated with the indicated primary and secondary antibodies. An ECL chemiluminescence solution kit (Meilunbio, Cat# MA0186, China) was used for immunoblot color development.

### 2.8. Measurement of Glutathione (GSH)

GSH is an important cellular antioxidant, and its oxidation type is GSSG. GSSG is reduced to GSH by glutathione reductase, and GSH is the main form in mammalian cells. GSH can react with the probe in the GSH test kit (Servicebio, Cat# G4305-48T, China) to produce a yellow product, and the absorbance of the reaction product at 412 nm can be measured to determine the GSH content. Briefly, cells seeded in the 6 cm dish were collected and resuspended with 400 μL PBS for ultrasonication at 40 W. Next, the ultrasonicated cells were centrifuged at 12,000 rpm at 4 °C for 15 min, then the supernatants were transferred into a new tube and mixed with deproteinizing reagent at the ratio of 1:1 followed by centrifugation at 12,000 rpm at 4 °C for 15 min. After centrifugation, the supernatants were collected and reacted with the probe for 5 min at 25 °C, the absorbance at 412 nm was detected by a microplate reader, and the data were analyzed by GraphPad Prism 9 software.

### 2.9. Statistical Analysis

Statistical analysis of the data was performed by GraphPad Prism 9. All quantitative data are presented as the mean ± SD of at least three independent experiments. Student’s *t* test was used for two-group comparison, and a one-way analysis of variance (ANOVA) was used for simultaneous comparison of more than two groups. *p* < 0.05 was considered statistically significant.

## 3. Results

### 3.1. Auraptene Exerts Anti-Tumor Effects in HCC Cells

Auraptene, a natural product, was reported to inhibit cell proliferation and invasion in some tumors, which include geranyloxyl at the C-7 position (Figure 1A). In order to explore whether auraptene has anti-tumor effects in HCC cells, we firstly treated HCCLM3 and HLE cells with different concentrations of auraptene and CCK-8 assay, which showed that auraptene inhibits HCC cell viability in a dose-dependent manner (Figure 1B). Colony formation also showed that auraptene impedes HCC cell growth (Figure 1C). Through microscopy observation, we found that auraptene can induce HCC cell death in a dose-dependent manner (Figure 1D). To further confirm whether auraptene has the ability to induce HCC cell death, HCC cells with different concentrations of auraptene were stained with PI for flow cytometry analysis. The results revealed that auraptene can significantly induce HCC cell death (Figure 1E). In summary, these results demonstrate that auraptene exerts anti-tumor effects in HCC cells via inducing cell growth inhibition and cell death.

### 3.2. ROS Induction Is Responsible for Auraptene-Induced Cell Growth Inhibition and Cell Death

It has been reported that auraptene can increase ROS levels in mouse colon cancer cell lines [33] and acute myeloid leukemia cell lines [30]. However, it is elusive whether ROS production is the main reason that auraptene exerts anti-tumor effects in HCC cells. We then analyzed the total cellular ROS levels after auraptene stimulation. The results showed that auraptene significantly increases the cellular ROS levels of HCCLM3 and HLE cells, and this upregulation was eradicated by the ROS scavenger N-acetyl-L-cysteine (NAC) and reduced L-glutathione (GSH) (Figure 2A). Importantly, the inhibitory effects of cell viability (Figure 2B) and colony formation (Figure 2C) induced by auraptene were diminished with the pretreatment of NAC or GSH. Next, we wanted to know whether ROS production is critical for auraptene-induced HCC cell death. The microscopic photographs (Figure 2D) and cell death analysis with flow cytometry (Figure 2E) showed that pretreatment with the ROS scavenger NAC and GSH impedes the cell death induced by auraptene in HCC cells. Taken together, these data suggest that auraptene-induced ROS production plays an important role in its anti-tumor effects in HCC cells.

### 3.3. Auraptene Induces HCC Cell Ferroptosis

Given our results indicating that auraptene induces HCC cell death via excessive ROS production, we wanted to examine in detail the exact form of cell death. Ferroptosis is a recently discovered programmed cell death characterized by ROS- and iron-dependent lipid peroxidation. Whether auraptene induces HCC cell ferroptosis is unknown. To answer this question, we used two ferroptosis inhibitors: ferrostatin-1 (Fer-1), and deferoxamine mesylate (DFO). A cellular total ROS-level analysis of HCCLM3 and HLE cells revealed that Fer-1 and DFO can significantly inhibit auraptene-mediated upregulation of ROS production (Figure 3A). Accumulation of lipid ROS is the key characteristic of ferroptosis, so we used the lipid ROS probe C11-BODIPY 581/591 to detect lipid ROS levels. The results showed that auraptene treatment can increase lipid ROS levels in HCCLM3 and HLE cells (Figure 3B). These data imply that auraptene can induce HCC cell ferroptosis. Next, we wondered whether the anti-tumor effects of auraptene in HCC cells are dependent on ferroptosis initiation. Colony formation (Figure 3C) and cell viability assay (Figure 3D) showed that Fer-1 or DFO treatment reversed the growth inhibition mediated by auraptene in HCC cells. More importantly, treatment with Fer-1 or DFO can inhibit HCC cell death induced by auraptene (Figure 3E,F). Overall, these results demonstrate that auraptene inhibits HCC cell growth and induces cell death via ferroptosis induction.

### 3.4. A Low Dose of Auraptene Sensitizes HCC Cells to Ferroptosis

The above data show that a high concentration of auraptene can directly induce HCC cell ferroptosis. A high concentration of auraptene may have side effects when used in clinical trials; therefore, we wanted to evaluate whether a low dose of auraptene can sensitize HCC cells to ferroptosis. The compound RSL3 (RAS-selective lethal small molecule 3) and cystine deprivation are two classical ferroptosis inducers. Therefore, we firstly used different concentrations (less than IC50 value) of an auraptene combination with RSL3 or cystine deprivation treatment to evaluate HCC cell growth inhibition. The colony formation assay showed that treatment with a low concentration of auraptene can increase the cell growth inhibition mediated by RSL3 or cystine deprivation (Figure 4A). Consistent with these results, treatment with a low concentration of auraptene can significantly increase RSL3- or cystine deprivation-induced HCC cell death (Figure 4B,D). Overall, these results reveal that a low auraptene concentration can sensitize HCC cells to ferroptosis, so a combination of auraptene with classical ferroptosis inducers may be a promising strategy for HCC treatment.

### 3.5. Auraptene Degrades SLC7A11

Ferroptosis initiation or defense is tightly controlled by some key proteins such as SLC7A11, GPX4, acyl-CoA synthetase long-chain family member 4 (ACSL4), and FSP1. To gain insights into the molecular mechanism of auraptene on HCC ferroptosis, we treated HCCLM3 and HLE cells with auraptene, and the Western blot showed that auraptene significantly decreases the expression of SLC7A11 instead of GPX4, ACSL4 and FSP1 (Figure 5A,B). In line with this finding, the ubiquitin–proteasome inhibitor MG132, but not the protein synthesis inhibitor chlorhexidine (CHX), eliminated downregulation of SLC7A11 protein levels induced by auraptene (Figure 5C). To determine whether the ubiquitin–proteasomal degradation of SLC7A11 was involved, we performed an immunoprecipitation assay, and the experimental data confirmed that auraptene treatment did promote the ubiquitination modification of SLC7A11 (Figure 5D). It is known that SLC7A11 plays an important role in the transport of cystine, and cystine was further used to synthesize GSH to inhibit peroxidation and ferroptosis. Therefore, we further detected the cellular GSH in HCC cells treated with auraptene. The results showed that auraptene treatment significantly decreased the cellular GSH levels of HCCLM3 and HLE cells (Figure 5E). These data demonstrate that auraptene targets SLC7A11 for ubiquitin–proteasomal degradation, implying that auraptene may initiate HCC ferroptosis via degrading SLC7A11.

## 4. Discussion

Ferroptosis, the non-apoptotic programmed cell death, was firstly discovered by Brent R. Stockwell in 2012 [37]. To promote growth, cancer cells exhibit higher iron requirements compared to normal cells in the body. This iron dependence makes cancer cells more susceptible to iron regulation and ferroptosis [38]. The liver is the main storage site for iron in the body. Since the liver acts as an important site for iron metabolism, ferroptosis plays an important role in HCC carcinogenesis [39]. Ferroptosis is initiated by excessive lipid peroxidation, which is produced by ROS and iron-dependent oxidation reaction. The SLC7A11-GPX4 axis is one of the important cellular ferroptosis defense pathways [40,41]. SLC7A11 is the cystine transporter which is critical for extracellular cystine uptake into cells [42]. Intracellular cystine can be metabolized to produce GSH, which is a cofactor of GPX4 that catalyzes the toxic lipid hydroperoxides (L-OOH) to non-toxic lipid alcohols (L-OH) [43], thereby eradicating ferroptosis. Recently, research about ferroptosis, especially in the field of tumor therapy, has attracted much attention [44]. The accumulation of ROS, induced by activated oxidative phosphorylation and upregulated lipid metabolism in tumor cells, makes them more susceptible to ferroptosis, so targeting ferroptosis may be a promising therapeutic strategy for cancer-specific treatment.

Auraptene is a natural organic compound with a molecular weight of 223 that can be extracted from citrus peel and has a characteristic orange fragrance and strong solvent properties. Many studies have reported that auraptene has anti-cancer activities. However, the detailed mechanism of its anti-tumor activity, especially in HCC, is largely elusive.

In this study, we focused on the effects of auraptene on HCC cells and found that auraptene induces HCC cell death via the ferroptosis pathway. Mechanistically, auraptene treatment inhibits SLC7A11 expression, and this inhibition can be eradicated by the addition of the ubiquitin–proteasome inhibitor MG132 instead of the protein synthesis inhibitor chlorhexidine (CHX). Moreover, auraptene treatment increases the poly-ubiquitination level of SLC7A11, implying that auraptene targets SLC7A11 for ubiquitin–proteasomal degradation in HCC cells. However, the specific ubiquitin E3 ligase or deubiquitinase involved in auraptene-induced SLC7A11 degradation needs to be explored in the future. Furthermore, SLC7A11 degradation leads to the decline of GSH and imbalance of cellular redox characterized by total ROS and lipid ROS accumulation, eventually resulting in ferroptosis and growth inhibition of HCC cells (Figure 6). Intriguingly, a low dose of auraptene can sensitize HCC cells to ferroptosis, implying the possibility of using auraptene alone or in combination with other ferroptosis inducers for cancer treatment. Although we found that auraptene targets SLC7A11 for ubiquitin–proteasomal degradation, the detailed molecular mechanism remains to be further studied.

## 5. Conclusions

Our study identified a new role for and the mechanism of the natural product auraptene in anti-tumor effects in HCC cells via ferroptosis induction. It provides a lead compound and therapeutic strategy using auraptene alone or in combination with other ferroptosis inducers for HCC treatment.

## Figures and Tables

**Figure 1 antioxidants-13-01015-f001:**
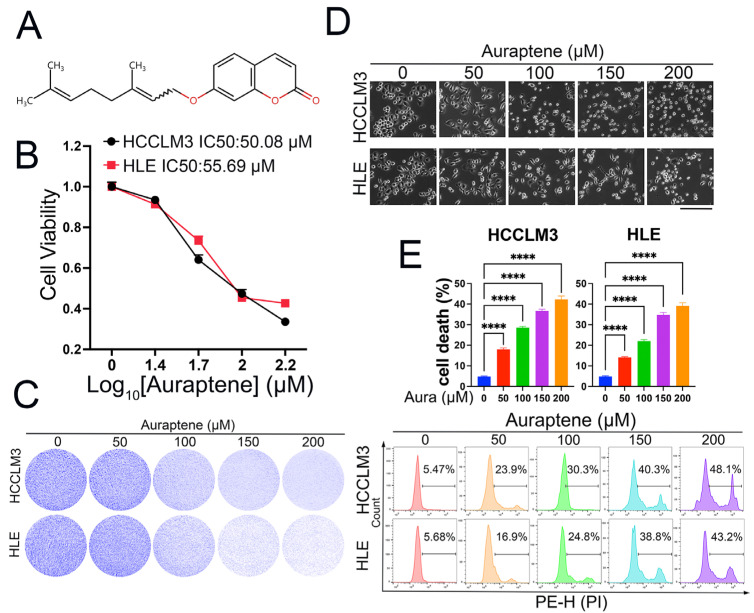
Auraptene exerts anti-tumor effects in HCC cells. (**A**) The molecular formula of auraptene with a molecular weight of 298.38. (**B**) HCCLM3 and HLE cells were plated into a 96-well plate at a density of 20,000 cells/well and treated with the indicated concentrations of auraptene for 24 h. Cell viability was detected with CCK-8 reagent and the IC50 was calculated. (**C**) HLE and HCCLM3 cells treated with the indicated concentrations of auraptene for 24 h were stained with crystal violet and photographed. (**D**) HLE and HCCLM3 cells treated with the indicated concentrations of auraptene for 16 h were photographed. Scale bar: 200 μm. (**E**) HLE and HCCLM3 cells treated with the indicated concentrations of auraptene for 16 h were stained with PI for flow cytometry analysis. Calculated cell death rate (Top) and representative pictures (Bottom) are shown. Aura: Auraptene. (**E**) Data are represented as the mean ± SD (n = 3), **** *p* < 0.0001 (one-way ANOVA).

**Figure 2 antioxidants-13-01015-f002:**
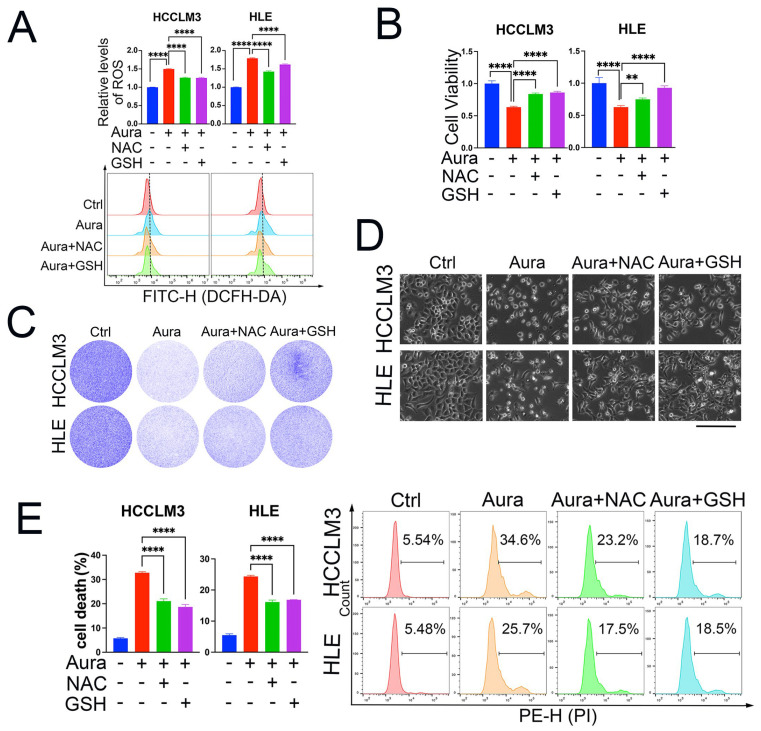
ROS induction is responsible for auraptene-induced cell growth inhibition and cell death. (**A**) HCCLM3 and HLE cells treated with or without 100 μM auraptene and 5 mM NAC or 5 mM GSH were stained with DCFH-DA for 1 h, followed by flow cytometry analysis. The calculated relative cellular ROS levels (Top) and histogram of flow cytometric pictures are shown (Bottom). (**B**) The cell viability of HCCLM3 and HLE cells treated with or without 100 μM auraptene, 5 mM NAC, or 5 mM GSH for 24 h was analyzed with CCK-8. (**C**) HCCLM3 and HLE cells treated with or without 100 μM auraptene, 5 mM NAC or 5 mM GSH for 24 h were stained with crystal violet and the photographs are shown. (**D**) HCCLM3 and HLE cells treated with or without 100 μM auraptene, 5 mM NAC, or 5 mM GSH for 16 h were photographed and the representative images are shown. Scale bar: 200 μm. (**E**) HCCLM3 and HLE cells treated with or without 100 μM auraptene, 5 mM NAC, or 5 mM GSH for 16 h were harvested and stained with 10 μg/mL PI followed by flow cytometry analysis. Calculated cell death rate (left) and representative flow cytometric pictures (right) are shown. (**A**,**B**,**E**) Data are represented as the mean ± SD (n = 3); ** *p* < 0.01, **** *p* < 0.0001 (one-way ANOVA).

**Figure 3 antioxidants-13-01015-f003:**
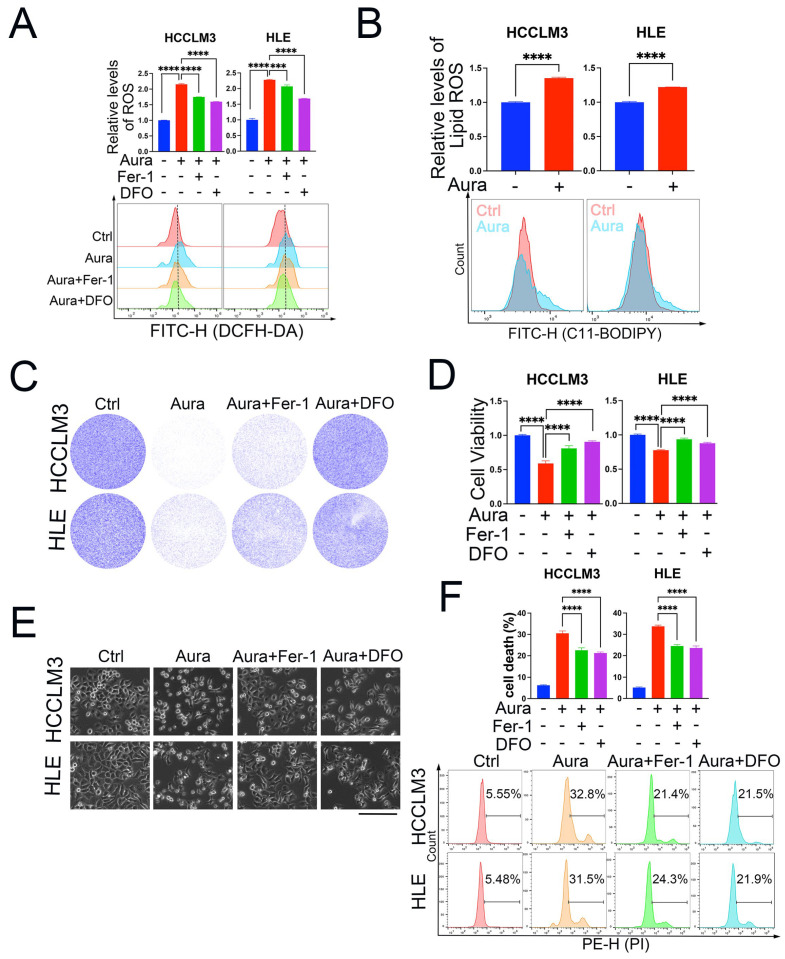
Auraptene induces HCC cell ferroptosis. (**A**) HCCLM3 and HLE cells treated with or without 100 μM auraptene, 2 μM Fer-1, or 50 μM DFO for 4 h were incubated with the ROS probe DCFH-DA for 1 h followed by flow cytometry analysis. The calculated total ROS levels (Top) and histogram of flow cytometric pictures (Bottom) are shown. (**B**) HCCLM3 and HLE cells treated with or without auraptene (100 μM) for 10 h were incubated with lipid ROS probe C11-BODIPY 581/591 for 1 h followed by flow cytometry analysis. The calculated lipid ROS levels (Top) and histogram of flow cytometric pictures (Bottom) are shown. (**C**) HCCLM3 and HLE cells treated with or without 100 μM auraptene, 2 μM Fer-1, or 50 μM DFO for 24 h were stained with crystal violet and photographed; the images are shown. (**D**) HCCLM3 and HLE cells treated with or without 100 μM auraptene, 2 μM Fer-1 or 50 μM DFO for 24 h were incubated with CCK-8 followed by an analysis with a microplate reader. (**E**,**F**) HCCLM3 and HLE cells treated with or without 100 μM auraptene, 2 μM Fer-1 or 50 μM DFO for 16 h were photographed. Scale bar: 200 μm. (**E**) or stained with PI followed by analysis with flow cytometry. (**F**) The calculated cell death rate (**F**, **Top**) and the representative flow cytometric pictures (**F**, **Bottom**) are shown. (**A**,**B**,**D**,**F**) Data are represented as the mean ± SD (n = 3); *** *p* < 0.001, **** *p* < 0.0001 (Unpaired Student’s *t* test for (**B**) and one-way ANOVA for (**A**,**D**,**F**)).

**Figure 4 antioxidants-13-01015-f004:**
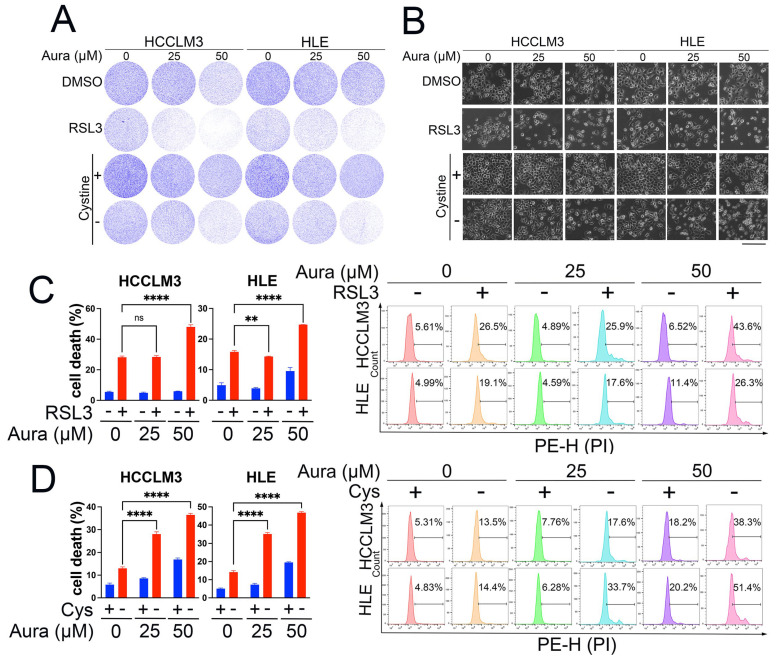
A low dose of auraptene sensitizes HCC cells to ferroptosis. (**A**) HCCLM3 and HLE cells treated with or without the indicated concentrations of auraptene, RSL3 (2 μM) for 24 h, or cystine deprivation for 36 h were stained with crystal violet and photographed; the images are shown. (**B**) HCCLM3 and HLE cells treated with or without the indicated concentrations of auraptene, RSL3 (2 μM) for 24 h, or cystine deprivation for 36 h were photographed and the representative images are shown. Scale bar: 200 μm. (**C**) HCCLM3 and HLE cells treated with or without indicated concentration of auraptene or RSL3 (2 μM) for 24 h were stained with PI followed by flow cytometry analysis. The calculated cell death rate (left) and the representative flow cytometric pictures (right) are shown. (**D**) HCCLM3 and HLE cells treated with or without the indicated concentrations of auraptene or cystine deprivation for 36 h were stained with PI followed by flow cytometry analysis. The calculated cell death rate (left) and the representative flow cytometric pictures (right) are shown. (**C**,**D**) Data are represented as the mean ± SD (n = 3); ns: no significance, ** *p* < 0.01, **** *p* < 0.0001 (one-way ANOVA).

**Figure 5 antioxidants-13-01015-f005:**
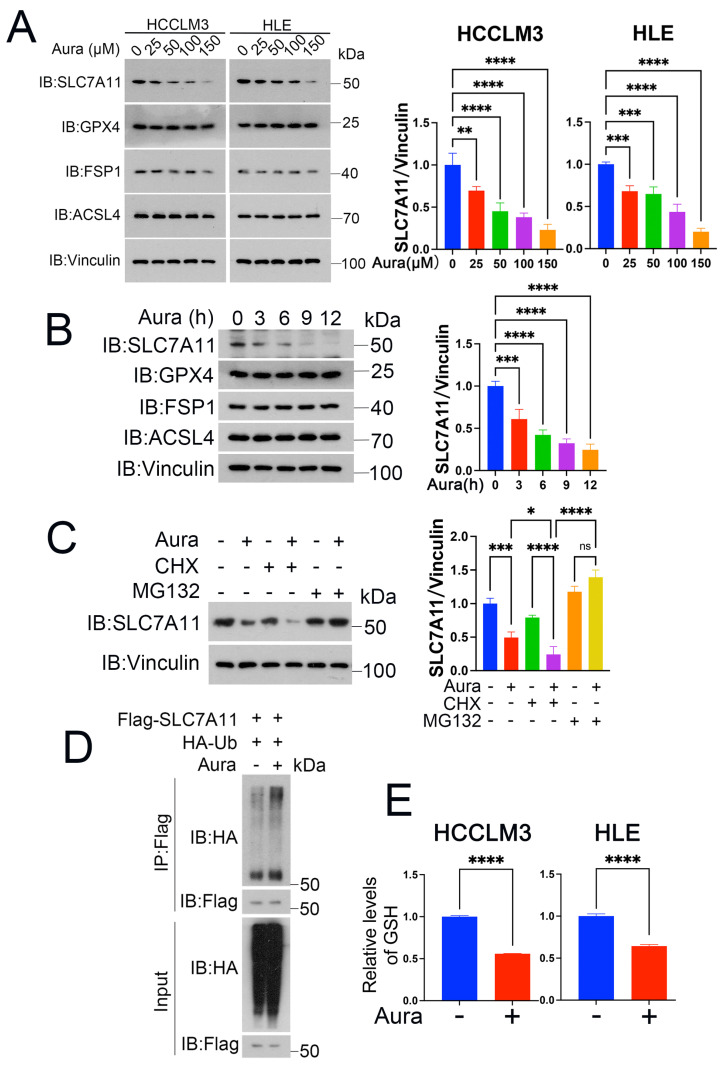
Auraptene degrades SLC7A11. (**A**) HCCLM3 and HLE cells treated with indicated concentrations of auraptene for 10 h were harvested for WB analysis with indicated antibodies, with Vinculin as the loading control. (**B**) HLE cells treated with 100 μM auraptene at the indicated time points were harvested for WB analysis with indicated antibodies, with Vinculin as the loading control. (**C**) HLE cells were pretreated with 100 μM auraptene for 2 h and then treated with or without CHX (100 μg/mL) or MG132 (10 μM) for another 8 h, then cells were harvested for WB analysis with indicated antibodies. (**A**–**C**) The data of SLC7A11/Vinculin are represented as the mean ± SD (n = 3); ns: no significance, * *p* < 0.05, ** *p* < 0.01, *** *p* < 0.001, **** *p* < 0.0001 (one-way ANOVA). (**D**) HLE cells were transfected with the specified plasmids for 15 h and then treated with or without 100 μM auraptene for 10 h. Cells were lysed for immunoprecipitated with anti-flag antibodies followed by Western blotting with the specified antibodies. (**E**) HCCLM3 and HLE cells were treated with or without 100 μM auraptene for 10 h and the cellular GSH levels were determined by a microplate reader at 412 nm. Data are represented as the mean ± SD (n = 3); **** *p* < 0.0001 (Unpaired Student’s *t* test).

**Figure 6 antioxidants-13-01015-f006:**
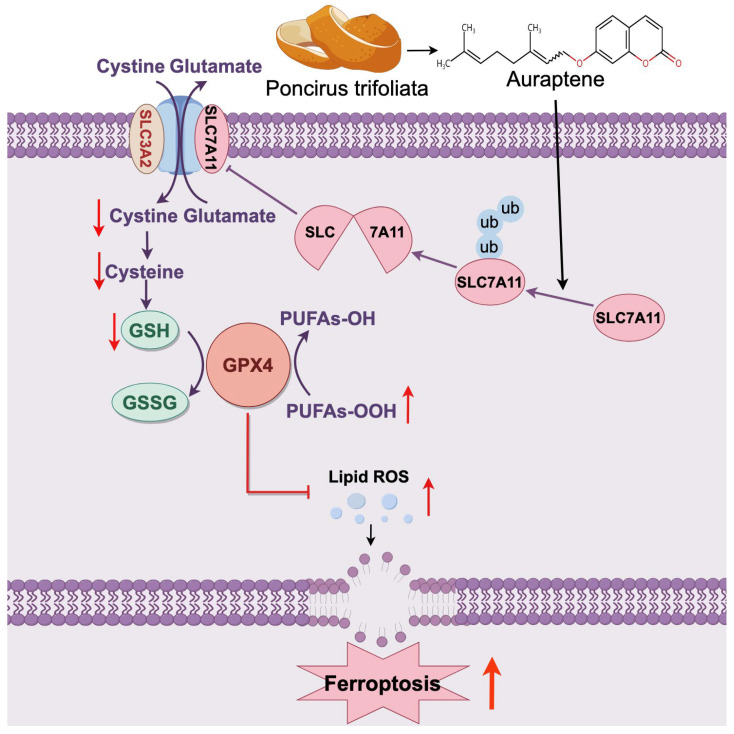
The working model of auraptene in HCC ferroptosis induction. Auraptene, the major coumarin of citrus plants, targets SLC7A11 for ubiquitin–proteasomal degradation, leading to lipid ROS production and ferroptosis of HCC. Ub: ubiquitin.

## Data Availability

The data presented in this study are available on request from the corresponding author.

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
