# Peer review of "Natural Product Auraptene Targets SLC7A11 for Degradation and Induces Hepatocellular Carcinoma Ferroptosis"

_antioxidants, 2024, doi:10.3390/antiox13081015_

Round 1

Reviewer 1 Report

I think this has been a careful study with multiple analyses. However, I have some concerns that in my view need to be addressed before the paper would be acceptable. 

1. There is a tendency of the authors to confuse the effects and interpretations between hepatocellular cancer and HCC cell lines. This needs to be clarified especially in the Abstract and Introduction.

2. SLC7A11 is proposed as a major player in the effects of auraptene and indeed this is portrayed in the title. However, the evidence in Figure 5 is weak.  I don't understand the difference in clarity of SLC7A11ab between  blot A and B/D. The latter do not give me the greatest of confidence for the interpretation of auraptene  targeting SLC7A11 for ubiquitin-proteasomal degradation.  Were the blots quantitated? It would be advisable to repeat them with a better antibody etc.

3. The Discussion needs to be expanded considerably. It is well known that human HCC tumors exclude iron but also some preneoplastic foci may actually show the reverse. Clearly this is likely to be a to be a complicated story.

4. Some aspects of the English require attention. I have annotated a copy of the paper with my particular comments.  

Please see attached copy with suggestions and questions.

Reviewer 2 Report

Natural product auraptene targets SLC7A11 for degradation  and induces hepatocellular carcinoma ferroptosis, finds that the long-studied compound seems to induce ferroptosis in HCC cells.  In general, the points seem to establish that ferroptosis is at least one component of the induction of cell death and provides some evidence that auraptene is involved in the degradation of SLC7A11, which could induce ferroptosis by depletion of GSH.  

I can never understand how authors can point out that a key enzyme in the process, GPX4, is a selenoprotein and not bother to supplement the culture media with selenium, given the tendency for 10% FBS/FCS to have suboptimal levels of selenium [Parant F, Mure F, Maurin J, Beauvilliers L, Chorfa C, El Jamali C, Ohlmann T, Chavatte L. Selenium Discrepancies in Fetal Bovine Serum: Impact on Cellular Selenoprotein Expression. Int J Mol Sci. 2024 Jul 1;25(13):7261. doi: 10.3390/ijms25137261. PMID: 39000368; PMCID: PMC11242189; Leist M, Raab B, Maurer S, Rösick U, Brigelius-Flohé R. Conventional cell culture media do not adequately supply cells with antioxidants and thus facilitate peroxide-induced genotoxicity. Free Radic Biol Med. 1996;21(3):297-306. doi: 10.1016/0891-5849(96)00045-7. PMID: 8855440]. One point that could be supplied to support the argument is to add a docking study to examine whether SLC7A11had a pocket that would bind auraptene or whether the induction of ubiquitination would involve some other mechanism.  Also, why is there no direct measurement of GSH and GSSH levels in the cells?

Line 54, supply a reference for this point. I am not familiar with the cell lines used, please supply  references for their isolation and any properties of relevance to the study.  On lines123-125, provide the basis for the lipid peroxidation/ROS measurement by the kit-what molecules are assayed and what is the indicator?  On line, 156, is the geranylaoxyl, portion the key structural feature in relation to ferroptosis?

Reviewer 3 Report

"The article entitled “ Natural product auraptene targets SLC7A11 for degradation 2 and induces hepatocellular carcinoma ferroptosis” is a well written article focused in evidencing the role of auraptene, specifically at different concentrations treatments, as modulator of cell viability mediated by ferroptosis. The possibility to evidence the role of natural products as adjuvants of chemotherapy is a very important topic in cancer research and for this reason this article has a significative relevance. As a minor revision : Fig 1 B should be modified reporting for IC50 analysis log of [auraptene] instead of [auraptene] The request to modify the analysis is relevant since only using log of [auraptene] instead of [auraptene] It is possible to obtain the correct IC50 value"

As a minor revision  : Fig 1 B should be modified reporting for IC50 analysis log of [auraptene] instead of [auraptene]

Round 2

Reviewer 1 Report

Manuscript now improved and blot problems addressed

I have no further comments after 1st review

Reviewer 2 Report

The authors addressed my commnets adequately.

None